# Physiological and clinical effects of low-intensity blood-flow restricted resistance exercise compared to standard rehabilitation in adults with knee osteoarthritis–Protocol for a randomized controlled trial

Brian Sørensen[1,2,3]*, Per Aagaard[1], Mikkel H. Hjortshøj[2,3,4], Sofie K. Hansen[5], Charlotte Suetta[5,6], Christian Couppé[2,3], S. Peter Magnusson[2,3], Finn E. Johannsen[2]

1 Department of Sports Science and Clinical Biomechanics, University of Southern Denmark, Odense M, Denmark, 2 Institute of Sports Medicine Copenhagen (ISMC), Bispebjerg Hospital, Copenhagen NV, Denmark, 3 Department of Physical & Occupational Therapy, Bispebjerg Hospital, Copenhagen NV, Denmark, 4 Centre for Health and Rehabilitation, University College Absalon, Slagelse, Denmark, 5 Department of Geriatric and Palliative Medicine, Bispebjerg and Frederiksberg Hospital, Copenhagen NV, Denmark, 6 Faculty of Health, Institute of Clinical Medicine, University of Copenhagen, Copenhagen, Denmark

* bsoerensen@health.sdu.dk

## Abstract

### Introduction

Osteoarthritis (OA) is a common disease with high socioeconomical costs. In Denmark, standard rehabilitation (SR) consists of a combination of patient education and supervised physical exercise involving a standardized neuromuscular training program. As an evidence-based alternative, high–load (>70% 1RM) resistance training (HIRT) has shown positive rehabilitation effects in knee-OA but may not be tolerated in all patients (~25%) due to knee joint pain. However, low-load resistance training (20–40% 1RM) with concurrent partial blood-flow restriction (BFR) appears to produce effects similar to HIRT yet involving reduced joint pain during and after exercise. The aim is to examine the effect of low-load BFR training compared to SR on pain, thigh muscle mass and muscle function in adults with knee-OA. We hypothesize that 12 weeks of BFR will lead to superior improvements in pain, muscle mass and mechanical muscle function compared to SR.

### Methods and analysis

90 participants diagnosed with radiographic knee-OA will be randomized to either BFR or SR twice a week for 12 weeks. BFR will consist of two selected lower limb strength exercises performed with an inflated pneumatic occlusion cuff. Intervention procedures in SR consist of a full 8 weeks GLA:D course followed by 4 weeks of team group training. Primary outcome variable is the change in KOOS-Pain subscale from baseline to 12 weeks. Secondary outcome variables are changes in pain sensitivity, functional performance, muscle mass and mechanical muscle function. Intention-to-treat and per-protocol analyses will be

relevant data from this study will be made available upon study completion.

**Funding:** This work was funded by: - The A.P. Moller Foundation (grant number 20-L-0186) (BS) (https://www.apmollerfonde.dk) - The Health Foundation (grant number 20-B-0214) (BS) (https://helsefonden.dk) - The Danish Rheumatism Association (grant number R181-A6356) (FEJ) (https://www.gigtforeningen.dk) - The Physiotherapy Practice Foundation (grant number R176-A4095) (BS) (https://www.fysio.dk/fafo/fonde/praksisfonden) - Foundation of the National Health Security System (Fonden for Faglig Udvikling af Speciallægepraksis) (Denmark) (grant number A2387) (FEJ) (https://rltn.dk/fonde/praksisfondene/fonden-for-faglig-udvikling-i-speciallaegepraksis) - The Aase and Ejnar Danielsen's Foundation (grant number 18-10-0559) (SKH) (https://danielsensfond.dk) The funders had no role in study design, data collection and analysis, decision to publish, or preparation of the manuscript.

**Competing interests:** The authors have declared that no competing interests exist.

conducted. One-way analysis of variance will be performed to evaluate between-group changes. Pre-to-post intervention comparisons will be analyzed using a mixed linear model. Regression analysis will be performed to evaluate potential associations between selected outcome variables.

## Introduction

Osteoarthritis (OA) is a highly common disease worldwide with more than 1 million Danish citizens (~20%) suffering from OA in one or multiple joints [1,2]. A significant portion of these individuals receive treatment to reduce pain and sustain work ability, leading to estimated socioeconomic costs related to OA of approximately 11 billion Danish kroner (1,5 billion EUR) per year [2].

Knee-OA is the most common OA-diagnosis and it is estimated that 60.000 Danish people with symptoms of knee-OA seek medical treatment every year [3]. The incidence of knee-OA is related to overweight, inactivity, aging, earlier knee injury, muscle weakness and exposure to lifelong physical work [4–6]. The increasing number of elderly as well as the growing increase in inactive and overweight people implies that the occurrence of knee-OA may be expected to increase in forthcoming decades.

Representing the most widespread non-medical and non-surgical treatment option both internationally and in Denmark, knee-OA patients are typically offered a combination of patient education, weight loss counseling and physical exercise [3,7]. Physical exercise including conventional heavy-resistance strength training, functional training (exercises handling own body weight) and cardiovascular training all seem to improve knee pain, functional function, and quality of life in people with OA [8–13]. In Denmark, the current paradigm of standard rehabilitation (SR) is termed GLA:D (Good Life with osteoArthritis in Denmark), which consists of eight weeks of supervised multi-component training performed twice weekly [14–16]. The concept is a combination of education and supervised neuromuscular exercises (typically abbreviated NEMEX) delivered by GLA:D-certified physiotherapists, with the purpose to increase joint range of motion (ROM), improve lower limb muscle strength and increase muscular stability around the knee- and hip joints, respectively [14–16]. Components of NEMEX have previously been demonstrated to yield positive changes in pain perception [11,15–20], functional capacity [11,15] and quality of life [15,16]. Nonetheless, the effect of NEMEX on lower limb muscle mass and strength remains unexplored. Notably, deficits in maximal muscle strength are often a critical factor in people with knee-OA, typically demonstrating muscle weakness of 20–40% compared to healthy sex and age-matched individuals [21–23]. Prior systematic reviews have indicated that reduced knee extensor muscle strength is a significant risk factor for an increased incidence of knee-OA while also disposing for increased severity of symptoms and accelerated decline in functional performance [6,24]. As such, improving lower limb muscle strength, with a particular focus on maximal knee extensor strength, may be a key factor in relieving symptoms and improving function in people affected by knee-OA. Those OA-patients who can tolerate heavy strength training, typically experience a positive effect on maximal muscle strength and power [25,26]. Unfortunately however, a substantial proportion of OA patients are unable to tolerate this type of training due to excessive joint pain during and following exercise sessions [27].

In recent years, strength training combined with concurrent blood flow restriction, often termed occlusion training, has gained increasing acceptance and usage in athletes [28,29] as

well as in different patient populations [30–38]. This type of training, often referred as BFR (Blood Flow Restricted) muscle exercise, typically is performed using low exercise loads (20–40% of MVC or 1RM) combined with restricted blood flow to the working muscles, the latter achieved by means of a pneumatic blood pressure cuff [39,40]. BFR training has been documented to result in significant improvements in muscle mass and muscle strength even with just a few weeks of intense daily training [41–44]. Moreover, of relevance especially in the clinical setting, BFR exercise has been reported to activate myogenic stem cells (satellite cells), which are involved in skeletal muscle regeneration and myofiber growth [42,45]. Notably, the improvements in muscle mass and strength with BFR training seem to be comparable or exceeding that achieved by conventional heavy-resistance strength training [31,34,37,38,41,46]. Recent data indicate that BFR exercise can have an acute pain-reducing effect [33,38] and result in greater strength gains and more pronounced reductions in pain with daily activities compared to heavy-resistance strength training in knee patients who experiences pain during training [30]. Based on these observations, BFR exercise may represent an attractive alternative training modality in patients with knee-OA.

The aim of the present study, therefore, is to investigate the effect of low-intensity BFR exercise on knee joint pain/function, muscle mass, and mechanical muscle function compared to SR in adults with knee-OA. We hypothesize that 12 weeks of BFR will lead to superior improvements in pain, muscle mass and mechanical muscle function compared to SR.

## Materials and methods

The RCT study is registered at ClinicalTrials.gov (Identifier: NCT05437770) and any changes to the protocol will be reported at this site.

### Study design

The present trial is designed as a two-armed, randomized, controlled trial following the Consolidated Standards of Reporting Trials (CONSORT) guidelines [47]. Assessment will be performed at baseline, after 8 weeks of training and at the end of the intervention period (12 weeks). Patient-reported questionnaires will be completed 24 weeks after the intervention period. Muscle biopsies (vastus lateralis) will be obtained from a subsample of 30 participants at baseline and again by the end of the intervention period (12 weeks). The overall study design is summarized in Fig 1.

### Blinding and randomization

The primary assessor (BS) conducting the pre-, mid- and post-testing of physical function, mechanical muscle function, ultrasonography and all statistical analysis will be blinded to participants' group allocation. In contrast, it is not possible for study participants and therapists conducting the SR and BFR-training to be blinded for group allocation.

After baseline assessment, participants will be randomized (1:1) to either SR or BFR-training (incl. education) using the Research Electronic Data Capture (REDCap) randomization system. Participants will be stratified for gender and BMI $\geq$ 27. Further, muscle biopsies will be included in the stratification for the volunteering subgroup of participants. Flowchart of participant allocation procedures is presented in Fig 2.

### Study participants

Inclusion will take place via the Institute of Sportsmedicine Copenhagen (ISMC), and the Department of Physical and Occupational Therapy at Bispebjerg Hospital, University of

| | Allocation | | Post allocation | | | | |
|---|---|---|---|---|---|---|---|
| | Screening | Baseline | 12 weeks of intervention | | | Primary endpoint | Follow-up |
| Week | -8 to 0 | 0 | 1 to 8 | 8 | 9 to 12 | 12 | 36 |
| **Enrollment** | | | | | | | |
| Eligibility criteria | • | | | | | | |
| Informed consent | • | | | | | | |
| Radiography | • | | | | | | |
| Randomization (n=90) | | • | | | | | |
| **Interventions** | | | | | | | |
| Standard rehabilitation (n=45) | | | ●━━━━━━━━━━━━━━━━━━━━━● | | | | |
| BFR (n=45) | | | ●━━━━━━━━━━━━━━━━━━━━━● | | | | |
| **Assessments** | | | | | | | |
| KOOS | | • | | • | | • | • |
| Oxford Knee Score | | • | | • | | • | • |
| Tegner Activity Level Scale | | • | | • | | • | • |
| 30-s Chair Stand Test | | • | | • | | • | |
| 4x10m Fast-Paced Walk | | • | | • | | • | |
| Stair Climb Test | | • | | • | | • | |
| MVC & RFD (KinCom) | | • | | • | | • | |
| Quadriceps CSA (Ultrasound) | | • | | • | | • | |
| Leg Muscle Power (Power-Rig) | | • | | • | | • | |
| Pain Threshold (Pain Algometer) | | • | | • | | • | |
| Muscle Biopsy (n=30) | | • | | | | • | |

**Fig 1. Schedule of enrollment, interventions, and assessments.**

Copenhagen. Assessment for inclusion is made after referral from general practitioners or after consultation (elaborated below) with chief physician (FEJ) at ISMC. Recruitment will also include advertising through local newspapers, posters in public libraries and postings on social media, as well as invitations to attend lectures with information about the study.

Participants will be invited to an initial examination by our specialist in rheumatology (FEJ). At the consultation a standard clinical assessment will be performed, and the participant will be examined for meeting the explicit inclusion or exclusion criteria of the study. Specific inclusion and exclusion criteria are listed in Table 1. For participants meeting the inclusion criteria, a document containing information about the study procedures will be handed out. Following receival of all relevant oral and written information and meeting the objective X-ray criteria for OA, an informed consent will be obtained from all participants. Randomization procedures will take place following baseline testing, which will be performed at Bispebjerg Hospital by a blinded (to group allocation) assessor. Subsequently, training intervention procedures will be initiated in successive batches of participants. First participant was recruited 29.06.2022.

## Intervention procedures

**Standard rehabilitation protocol.** Each participant randomized to the SR group will be offered to participate in the current Danish standard rehabilitation (SR) protocol (termed GLA:D [16]. Specifically, GLA:D comprise a registered 8-weeks structured treatment program for people with symptomatic knee and hip OA, which includes 16 supervised group-based exercise sessions including 2–3 patient education sessions delivered by certified health care practitioners, most commonly physiotherapists [16]. Two patient education sessions will be delivered by the treating clinician(s) and, when possible, a third session will be delivered by an expert patient. This protocol is considered a minimum intervention package and contains individualization of the exercise program and additional treatments as deemed by the clinician [14–17,48]. In the present trial, the SR

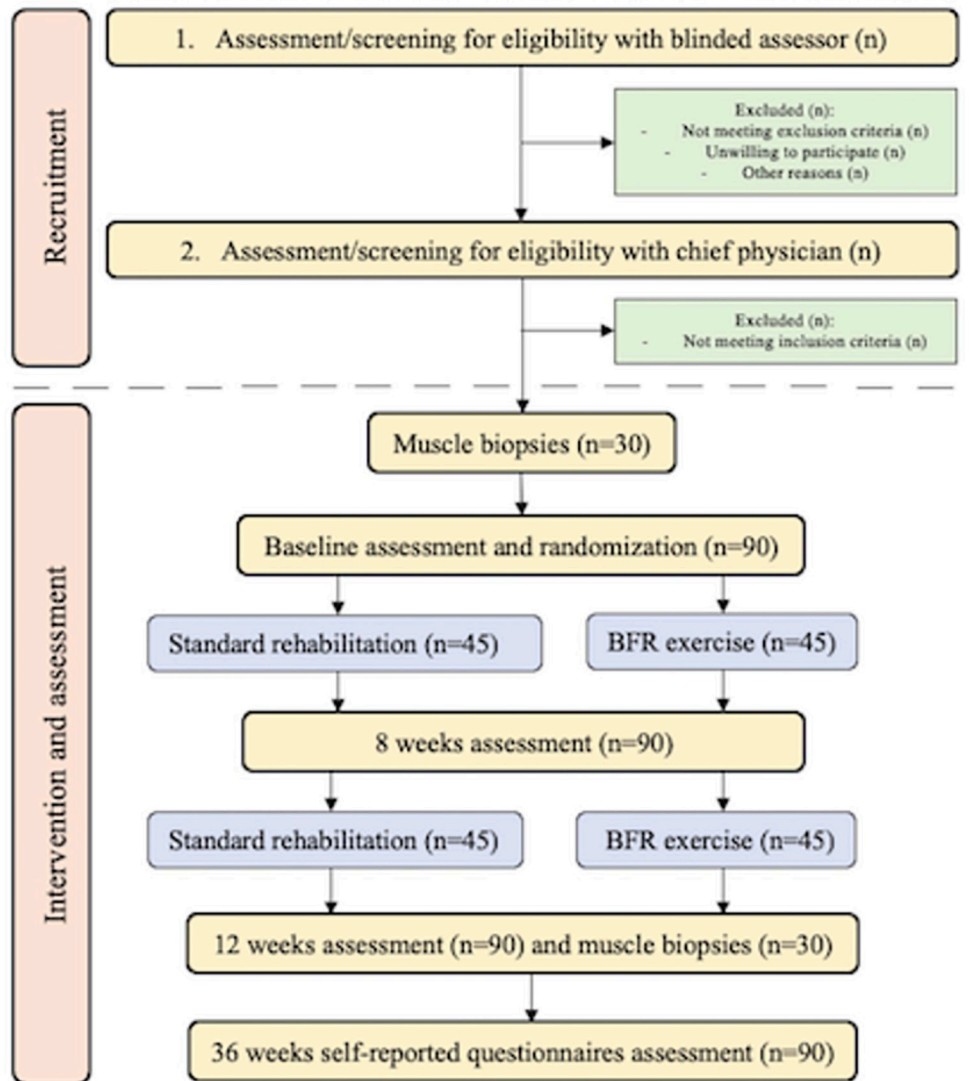

**Fig 2. Flowchart of participants.**

program is extended for 4 additional weeks where the participants continue the supervised team group training at the PT clinic twice a week. Training exercises in the additional 4 weeks will involve the same exercises as performed during the first 8 weeks. Follow-up testing will be conducted after 8- and 12-weeks training, respectively (more details given below).

The training sessions consist of three parts, which are all supervised by a trained physiotherapist: warming up, a circuit program, and cooling down. The warm-up comprise ergometer cycling for 10 minutes. The circuit program comprises four exercise circles, including neuromuscular exercises with key elements: core stability/postural function; postural orientation; lower extremity muscle strength; and functional capacity exercises. The exercises are mainly performed in closed kinetic chains. Because muscle weakness of the lower extremity, particularly the quadriceps, is common in patients with OA, open kinetic chain exercises are also performed to improve muscle strength of the knee and hip muscles. Two to six exercises are performed in each exercise circle. Each exercise is performed 2–3 sets of 10–15 repetitions (reps), with rest corresponding to one set (~60 s), between each set and exercise. The exercises

**Table 1. Inclusion and exclusion criteria.**

| Inclusion: | Exclusion: |
|---|---|
| • All participants must meet the American College of Rheumatology (ACR) criteria for OA<br>• Visible OA on X-ray pictures (Kellgren & Lawrence grade 2–3)<br>• Unilateral pain and functional limited for a minimum of 3 months<br>• Able to voluntarily (i.e. unassisted) perform a 90 degrees flexion in the knee<br>• Able to perform the machine exercises (knee extension and leg press) planned for the BFR exercise<br>• Danish-speaking<br>• No longer travel planned within the intervention period | • Kellgren & Lawrence grade 1 and 4<br>• Bilateral OA-symptoms<br>• Prior knee- or hip alloplasty<br>• Glucocorticosteroid injection in the knee within the last 6 months<br>• Inflammatory arthritis<br>• Known neurotic disease such as multiple sclerosis or peripheral neuropathy<br>• Prior myocardial infarct or apoplexy, or chest pain during physical activity<br>• Diabetes 1<br>• Other health related or medical conditions which makes it impossible participate in the study |
| **Exclusion criteria related to pneumatic occlusion training:** | |
| • Peripheral vascular disease<br>• Excessive varicose veins<br>• Prior history of deep venous thrombosis<br>• Venous insufficiens causing edema in the lower legs<br>• Systolic blood pressure over 160 mmHg or under 100 mmHg | |

are performed both bilaterally and unilaterally involving both the non-affected and the affected leg, although main focus is on the affected leg. To allow for progression, three levels of difficulty are given for each exercise. Progression is made when an exercise is performed with good sensorimotor control and good quality of the performance (based on visual inspection by the physiotherapist) and with minimal exertion and control of the movement (perceived by the patient). The last part of the training program includes cool-down stretching exercises for the lower extremity muscles (10 minutes) [8,49].

**BFR (Blood-Flow Restricted) exercise protocol.** Participants randomized to BFR resistance exercise will attend 24 supervised BFR sessions (two times a week for 12 weeks) delivered by experienced clinical BFR instructors at Bispebjerg Hospital. Each BFR session will consist of warm-up on an ergometer bike (10 min). This is followed by two different unilateral lower limb resistance exercises: (1) leg press and (2) knee extension performed in standard strength training machines (TechnoGym). Each exercise will be performed with the knee-OA diagnosed leg first followed by the contra-lateral leg, comprising four sets with a 30 second rest in between. 30 reps in the first set; 15 reps in second and third sets; and reps performed until exhaustion in the fourth set. If the participant can perform more than 15 reps in the fourth set, the exercise load will be increased 5% in the subsequent training session [50]. Participants will be instructed to perform both the eccentric and concentric contraction phases using a 2-s pace duration. The fourth set will be performed to the point of exhaustion defined as being unable to complete the final concentric contraction phase in 2 seconds. During the 30-s rest period in between sets, participants will rest in a self-chosen body position while maintaining the cuff-pressure. Between exercises (1) and (2), participants will receive a 5 min rest period with the cuff released [51]. Cuff pressure will be released immediately after completion of the final exercise round.

The occlusion pressure during both exercises will be set at 60% of the individual arterial occlusion pressure (AOP) in the first two training sessions, and increased to 70% AOP in the subsequent two sessions, to reach 80% AOP from the fifth training session, where it will remain during all subsequent training sessions [40,52–55]. The stepwise increments in cuff-pressure are chosen to habituate participants to the 80% AOP pressure. If a participant is

unable to perform training at 80% of AOP due to intolerable pain, the cuff-pressure will be decreased to 70% of AOP. Individual AOP will be determined using a pneumatic, conically shaped, 12-cm wide, pneumatic cuff (Occlude Aps, Denmark) attached to the participant's proximal thigh area on the knee-OA diagnosed side, and subsequently applied to the proximal thigh of the contralateral side. While sitting on an examination table with the ankle and 1/3 of the lower limb off the table, a vascular Doppler probe (EDAN Instruments, China) will be placed posterior to the medial malleolus over the posterior tibial artery to capture the auscultatory pulse. To determine the cuff pressure (mmHg) needed for total blood flow occlusion (arterial occlusion pressure: AOP), the cuff will gradually be inflated in successive 20-mmHg steps until reaching the pressure where the auscultatory pulse is interrupted (i.e., AOP). The first time the auscultatory pulse is interrupted, the examiner releases pressure from the cuff until the auscultatory pulse is present again. When the auscultatory pulse reappears, the cuff is inflated until AOP is identified again. If the second AOP is identical to the first, it will be defined as the AOP for that specific participant [51]. In case of more than 10-mmHg difference between the first and second measurements, a third will be conducted to determine AOP.

Initial load intensity during the BFR exercises will be 30% of 1 repetition maximum (1RM) in all exercises and for both limbs [53–55]. The initial training load will be estimated based on unilateral sub-maximal testing (5-10RM) for the knee-OA diagnosed leg first only prior to the first training session [56,57]. 1RM load intensity will be estimated from 5-10RM [57], and will be re-assessed at weeks 5 and 9 to make sure that 30% 1RM load intensity is adjusted to account for the progressive increase in 1RM strength throughout the intervention period.

**Patient education.** Patient education will consist of one to two lecture sessions (approximate total duration: 2 hours) given by a certified physiotherapist familiar to SR. The sessions intend to provide the patient with knowledge of OA and treatment of OA with a particular focus on exercise. Further, the beneficial effects of exercise on symptoms and general health will be discussed, and self-help advice will be offered. Finally, the patient education sessions will focus on engaging the patients actively to share experiences with each other [16].

Both intervention groups will attend the patient education sessions. Participants in the SR group will attend the education session at the specific physiotherapy clinic at which they train. Participants in the BFR group will attend the session at Bispebjerg Hospital performed by internal staff.

## Outcome variables

All study outcome variables are summarized in Table 2. Outcome assessment will be performed at baseline (prior to randomization and before intervention onset), and again after 8 weeks of training (the normal duration of SR) and 12 weeks of training, respectively. In addition, patient-reported questionnaires will be completed 24 weeks after the intervention period. One assessor (BS) blinded to group allocation will perform all baseline and follow-up testing. Prior to baseline testing, the assessor will be thoroughly trained in performing the tests according to the specific procedures involved in each test modality (elaborated below). Further, the assessor will be trained in how to communicate with the participants at follow-up test sessions to avoid unblinding due to miscommunication. Also, all cases where blinding is being broken will be registered.

The instructors in charge of the BFR resistance exercise procedures are therapists who are trained in performing this type of exercise in healthy subjects as well as in selected patient groups incl. OA patients. Prior to all follow-up test sessions, the physiotherapists and instructors in charge of both SR, and BFR will carefully remind the participants not to reveal their group allocation to the assessor at any point during the post-testing activities.

**Table 2. Summary of outcome variables.**

|  | Instrument for data collection |
|---|---|
| **Primary outcome** |  |
| Pain | KOOS-Pain subscale |
| **Secondary outcomes** |  |
| Symptoms, activities of daily living and quality of life | KOOS |
| Pain and function | Oxford Knee Score (OKS) |
| Activity level | Tegner Activity Scale |
| Functional performance |  |
| Lower body strength and endurance | 30-s Chair Stand Test |
| Lower body endurance and balance | 4x10m Fast-paced Walk Test |
| Lower body strength and balance | Stair Climb Test |
| Mechanical muscle function |  |
| Lower body muscle strength (MVIC) | KinCom, Isokinetic dynamometer |
| Lower body explosive muscle strength (RFD) | KinCom, Isokinetic dynamometer |
| Muscle Cross-Sectional Area (CSA) | Ultrasonography |
| Leg extensor muscle power | Power-Rig |
| Myocellular components | Muscle biopsy |
| Pressure Pain Threshold | Handheld pain algometer |

## Primary outcome

Knee injury and Osteoarthritis Outcome Score:

The Knee injury and Osteoarthritis Outcome Score (KOOS) is a patient-administered knee-specific questionnaire and is comprised of five subscales: Pain; Symptoms; Activities of daily living; Sport and Recreation, and Knee-Related Quality of Life. Each item is scored from 0 to 4 [58–61]. Primary outcome variable will be the between-group difference (0 to 12 weeks) in KOOS-Pain. KOOS-Pain consists of 9 questions regarding the difficulties the patient experience with physical activity due to their knee problems and is scored on a scale, ranging from zero (no problems) to four (extreme problems) for each question [59,60]. A normalized score for the entire subscale will be calculated and reported, ranging from zero (extreme symptoms) to 100 (no symptoms) [59,60]. Acceptable reliability and construct validity data have previously been reported for this variable [59].

## Secondary outcomes

The remaining four KOOS subscales (symptoms, activities of daily living, sport and recreation, and knee-related quality of life) also will be assessed, using the same approach as for the primary outcome variable.

Oxford Knee Score:

The Oxford Knee Score (OKS) is a patient-reported questionnaire comprising 12 questions about selected activities of daily living that reflects the patient's subjective assessment of their knee-related health status and benefits of treatment [58,62]. OKS has been developed and validated specifically to assess knee joint function and pain after total knee replacement and knee-OA [58,62].

## Secondary outcomes related to functional performance

30-second Chair Stand Test:

The 30-s chair stand test (30-s CST) is used to assess functional leg extensor strength, power and endurance. The 30-s CST will be performed using a rigid chair (seat height: 43–44 cm). The 30-s CST measures the number of sit-to-stand repetitions completed within 30-s.

Further, the 30-s test outcome will be converted to mean leg extensor power output using algorithms recently validated across the adult age span [63,64]. The 30-s CST is a valid and sensitive measure of lower-extremity sit-to-stand function with good-to-excellent intra- and interobserver reliability [65–68].

<u>4x10 m Fast-paced Walk Test:</u>

The 40-m fast-paced Walk test (40m-FWT) is a test of walking speed over short distances and changing direction during walking. It measures the total time it takes to walk 4 * 10 m excluding turns (m/s) [65,68]. Participants will be instructed to walk as quickly and as safely as possible without running to a visible mark 10 meters away, return and repeat for a total distance of 40 m [65,68]. Prior to the test, one practice trial will be provided to check understanding. The 40m-FWT is considered a valid and sensitive measure for assessing short distance maximum walking speed with excellent reliability [68].

<u>Stair Climb Test:</u>

The measurement of timed stair-climb performance has been employed extensively in the clinic and literature to measure efficacy of treatment in knee-OA patients [63]. The stair-climb test (SCT) is a test of lower body strength and balance [65,69–72]. This is a function that patients with lower limb OA find particularly painful and the ability to climb stairs has been strongly correlated with leg extensor power and joint flexibility [70].

The SCT involves ascending and descending 10 stairs measuring 18 cm rise / 92 cm width. Each participant is asked to ascend/descend the stairs at their maximal pace (without resting) [72]. The total time of stair ascend and descend is recorded electronically, with each participant performing a single trial after the familiarization trial. Use of a walking aid or the handrail will be noted [70,71].

## Secondary outcomes related to mechanical muscle function

<u>Maximal Voluntary Isometric Contraction strength and Rate of Force Development:</u>

Maximal Voluntary Isometric Contraction strength (MVIC) will be obtained in an isokinetic dynamometer (KinCom; Kinetic Communicator, Chattecx, Chattanooga, TN) as the maximal isometric knee extensor torque generated at 70˚ knee joint angle (0˚ = full extension) [73–75]. The reliability and validity of this setup have been described elsewhere [76]. Briefly, participants are seated 10˚ reclined and firmly strapped at the proximal part of the thigh. The axis of rotation of the dynamometer lever arm is visually aligned to the axis of the lateral femur condyle of the subject, and the lower leg is attached to the lever arm of the dynamometer just above the medial malleolus. Individual settings of the seat, backrest, dynamometer head, and lever arm length will be registered, so identical positioning can be achieved during follow-up testing. To correct for the effect of gravity on the recorded knee extensor torque, the passive mass (flexor torque) of the lower leg will be measured in the dynamometer at a knee joint angle of 45˚ [73]. Maximal isometric quadriceps contractions will be performed during static knee extension at a knee joint angle of 70˚ (0˚ = full knee extension) as described previously [74]. After 10-min of bicycle warmup followed by a number of submaximal preconditioning trials with increasing percentage of maximal contraction (~50–90% of maximal contraction), each participant performs three to four maximal contractions of the knee extensors. Participants are carefully instructed to contract "as fast and forcefully as possible" and instructed during testing to have their arms crossed over the chest to avoid compensation. On-line visual feedback of the instantaneous dynamometer force is provided to the participants on a computer screen. Trials with an initial countermovement (identified by a visible drop in the force signal) will be always disqualified, and a new trial will be performed. The trial with the highest maximal voluntary knee extensor peak torque is selected for further analyses [75].

Contractile RFD will be calculated as the average slope of the rising phase of the torque-time curve (i.e. RFD = ΔTorque/ΔTime) at 30, 50, 100, and 200 ms relative to onset of contraction (t = 0). Onset of contraction is defined as the instant where force increase 3.5 Nm above the rising baseline level, corresponding to ~2% of the peak isometric torque. Contractile impulse is measured as the area under the force-time curve ($\int$force d$t$) in the same time intervals [75]. Relative RFD is determined as the slope of the moment-time curve normalized relative to peak isometric torque, MVIC [73,75,77]. MVIC and RFD testing will be conducted on the symptomatic leg only.

Muscle Cross-Sectional Area:

A portable B-mode ultrasound (US) device (GE LOGIQ[TM] E10, GE Healthcare, USA) with linear-array probe (variable frequency band 4.2–13.0 MHz) will be used for measurements of muscle cross-sectional area (CSA). GE Logiq E10 LogicView[TM] software is used to generate panoramic axial CSA images from the quadriceps muscle (rectus femoris and vastus lateralis) [78,79].

Participants will be positioned supine with their legs extended and relaxed for 10 min to restore the normal flow of body fluids [80]. Orientated in the axial-plane, the US probe is positioned perpendicularly, and a water-based gel is used to promote acoustic contact between the skin and the probe [80]. The probe is moved manually with a slow and continuous movement from the medial to the lateral part of quadriceps along a marked line on the skin [78]. Great care will be taken to be consistent in applying minimal pressure from the US probe to the skin during all scannings to avoid compression (deformation) of the underlying muscle tissue. The anatomical site for all measurements will be at 50% of the distance between the lateral condyle and greater trochanter of the femur [83]. Transparency film will be used to map the skin to ensure that CSA measurements are matched between test-days [79]. After US scanning, CSA images will be reviewed from the monitor to ensure the preliminary quality of the images. Subsequently, three ultrasound panoramic CSA images will be obtained for later analysis. All images will be saved and exported for analyses in ImageJ software (Version 1.48v, National Institutes of Health, Bethesda, MD, USA). Quadriceps ultrasound imaging has previously been established as a valid and reliable measurement tool for assessing thigh muscle size (CSA) and quality [78,79,81–85].

Leg extensor power:

Maximal leg extensor muscle power (LEP) will be assessed using the leg extensor power-rig (University of Nottingham Medical School, Queen's Medical Centre, Nottingham NG7 2UH, United Kingdom) [11,86]. Participants will be seated in the power-rig chair and after 2–3 warm-up trials instructed to press (unilaterally) as hard and fast as possible onto a footplate connected to an instrumented flywheel system [87]. Visual feedback of the instantaneous power-time curve is provided on a computer screen after each trial and the participant will perform successive trials (30-s pause) until unable to increase mean power any further. The trial with highest power is selected for further analysis. The power output normalized to kilo body mass (Watt/kg) is calculated for the symptomatic leg and used for within-group and between-group analyses [11].

Muscle biopsy sampling:

Muscle biopsies will be obtained from the knee extensors (vastus lateralis: VL) for determination of selected myocellular components (fiber type composition and area, vascularization, satellite cell content, myonuclei number) [42,88,89]. In brief, muscle biopsies will be obtained (100–150 mg) from 30 volunteering participants (15 participants from each intervention group). The biopsies will be obtained unilaterally from the middle portion of the vastus lateralis muscle using the percutaneous needle biopsy technique of Bergström [90–92]. Biopsies will

be performed by experienced orthopaedic surgeons (chief physicians) trained in performing the needle muscle biopsy technique at Bispebjerg Hospital.

<u>Pressure Pain Threshold:</u>

The Pressure Pain Threshold (PPT) will be assessed using a handheld pain algometer (Algometer Type II; Somedic AB, Sollentuna, Sweden) using a 1 cm$^2$ probe. The participant is placed in a sitting position with the knees at 90° flexion [93,94]. The test instructor locates (palpation) the most painful area (MPA) in the medial knee joint line and two reference points. The muscle belly of the tibialis anterior is used as a reference point and located 3 cm distally and 2 cm laterally from the tuberositas tibia, and the muscle belly of extensor carpi radialis 3 cm distally from lateral epicondyle of the humerus. The test sides are then marked on a sheet picturing the knee and with a removable pen in order to replicate test sides. At post-testing sessions (week 8 and 12) both the MPA from the first test session and the MPA on the day of post-testing are being located and tested. The probe is placed perpendicular to the skin at the marked sides and pressure is applied orthogonally to the skin at a standardized and constant rate of application (30 KPa/s) [94]. The participant is provided with a hand-held button and will be instructed to press the button at the first instance the sensation change from pressure to pain. When the participant presses the button, the algometer will record the applied pressure (expressed in KPa), which will be recorded as the PPT. The mean of two consecutive test scores will be calculated and recorded as the PPT score for each test side with a 1-minut rest period between tests [93]. The participant will be assessed at rest unilaterally in both limbs, where PPT on the knee-OA diagnosed leg will be measured first. In order to prevent adverse effects (e.g., soft tissue damage) maximum pressure will never exceed 1000 KPa. The pain algometer has proven valid and reliable in patients suffering from knee OA [95–98].

<u>1RM leg press strength:</u>

1RM leg press strength will be estimated from a 5-10RM leg press test [57]. Participants will perform three warm-up sets at submaximal loads (~25–50% 1RM). The first and second warm-up sets consist of 12 repetitions, and the third warm-up set consists of eight repetitions. After warm-up, loads will be increased to determine 5-10RM [56]. 1RM strength will be estimated from the 5-10RM values using correlational equations reported elsewhere [56,57]. 1RM strength testing will only be performed by participants in the BFR group to estimate the initial training load.

<u>1RM knee extensor strength:</u>

1RM knee extensor strength will be estimated as described above for the 1RM leg press test [56,57].

## Ethical aspects and dissemination

The study is approved by the Committees on Health Research Ethics in Region Hovedstaden (H-19079135). The study will be carried out in accordance with international, standardized research ethics considerations and has been approved by the Danish Data Protection Agency (P-2019-814).

All included experimental methods have previously been used at Bispebjerg Hospital (Institute of Sports Medicine and Geriatric Research Unit), Herlev Hospital (Geriatric Research Unit, Department of Internal Medicine) and University of Southern Denmark (Research Unit of Muscle Physiology and Biomechanics, Department of Sport and Biomechanics).

There is no indications in the literature that patients with knee-OA are unable to tolerate or complete 12 weeks of BFR-training [25,26], and therefore it was deemed that the low exercise loads employed with the present BFR resistance training protocol coupled with the controlled movements performed in the exercise machines should be tolerable to the present group of

OA patients. In support of this notion, OA patients typically demonstrate a high training compliance to low-load BFR resistance training [25,38].

Prior to study inclusion, all participants will provide written informed consent in accordance with the Declaration of Helsinki. All data and information collected in the trial will be treated blinded and encrypted to the researchers and staff connected with the trial.

**Patient and public involvement.** A pilot trial was performed before developing this clinical trial to determine the efficacy and feasibility of BFR in a representative adult suffering from bilateral symptomatic knee-OA. The feedback from the participant on the specific training protocol regarding training intensity, frequency and duration was useful for optimizing the BFR exercise protocol used in the present trial.

## Data management

All data from the patient-reported outcome measures (KOOS, OKS, Tegner score), functional performance tests and mechanical muscle function tests will be entered into RedCap by the blinded assessor. All patient data will be anonymised by assigning study numbers (FP01, FP02. . .) to each patient. The raw data will be stored for five years after completion of the trial with restricted access to the data. After publication of the trial, a fully anonymised patient-level dataset and corresponding statistical

description will be made publicly available if required by the scientific journal, in which the results are published.

## Statistical considerations

**Sample size.** The estimated number of participants in the study are based on the primary outcome variable, KOOS-Pain subscale [58], with the assumption that a between-group change of 10 points is considered clinically relevant [58,60] assuming a standard deviation (SD) of 15 points [11,60]. To achieve a statistical power of 80% with a significance level of 0.05 and an expected between-group change of 10-points following 12 weeks training (primary endpoint) using a two-sample pooled $t$ test required 37 participants in each group. To compensate for potential dropouts, a total of 90 participants are planned to be included in the study randomly assigned to the two intervention groups.

**Statistical analysis.** The change in the primary outcome (KOOS-Pain) from baseline to primary endpoint (12 weeks) will be calculated for both groups, and mixed linear model analysis with an autoregressive covariance structure will be used to examine if there is a systematic difference between the two intervention groups. Both intention-to-treat analysis (i.e., including all randomized participants independent of departures from allocation treatment, compliance and/or withdrawals) and per-protocol analysis (participation in ≥80% of all scheduled training sessions) will be conducted.

All outcome parameters (primary, secondary) will be analyzed using a one-way analysis of variance model to analyze between-group mean changes. The model includes changes from baseline to 8- and 12-weeks. Pre-to-post intervention comparisons will be analyzed using a mixed linear model approach. Explorative analysis will also be conducted. The primary outcome will be correlated to the various secondary outcomes using multiple linear regression analyses. The level of statistical significance will be set to p<0.05 (two-tailed testing). All statistical analyses will be performed in a blinded fashion SPSS (Version 22, IBM SPSS Statistics).

## Discussion

To our best knowledge, this is the first clinical trial to investigate the effect of blood-flow restricted BFR strength training vs. standard rehabilitation (SR) on pain intensity, functional capacity, and mechanical lower limb muscle function in adults with knee OA. In Denmark,

most hip- and knee OA patients are referred from their physician to physiotherapy-based SR involving participation in the GLA:D program. The GLA:D program is a Danish initiative started in 2013 that aims at facilitating evidence-based care of patients with OA, with key components consisting of patient education and neuromuscular exercise therapy delivered by GLA:D certified physiotherapists [16,99]. The GLA:D program is currently available for patients in Denmark, Canada, Australia, China, and Switzerland [14]. Previous studies have investigated the effects of SR on musculoskeletal pain and functional performance in OA patients [14,16,18,20]. Two studies have investigated the effect of conventional (i.e. free-flow) low-load strength training with SR [11,19] and a single study has compared SR to intra-articular saline injections [17]. Interestingly, the addition of low-load conventional strength training to SR appeared to provide no additional benefits on self-reported physical function or mechanical muscle function, except for an improved stair climb performance and reduced pain sensitization [11,19]. Further, comparing SR to intra-articular saline injections revealed comparable improvements in various symptomatic and functional outcome measures [17]. However, to our best knowledge, no previous study has examined the effect of low-load BFR training compared to SR on lower limb mechanical muscle function, self-reported physical function, and objective functional performance in patients with lower limb (knee) OA.

Despite that quadriceps muscle weakness is associated with increased risk of early onset and accelerated progression of knee-OA [4,6,22,100], the effects of SR on maximal quadriceps muscle strength have not yet been examined. Only a single study seems to have investigated the effect of SR intervention on mechanical lower limb muscle function [11]. In that study, half of the participants were randomized to conventional low-load strength training approximately 10 min after a preceding SR session twice weekly for 12 weeks. Maximal leg extensor power was assessed using a Nottingham Power-Rig and remained unchanged both with SR alone and when combined with conventional low-load resistance training [11].

The effect of low-intensity BFR training has been investigated in various patient groups, including knee OA patients [25,26,101]. A systematic review and meta-analysis compared the effect of low-intensity BFR training to low-intensity (LIRT), moderate-intensity (MIRT) and high-intensity (HIRT) conventional (i.e. free-flow) resistance training on maximal muscle strength, muscle mass and functional performance in patients with knee OA [37]. No differences were observed between BFR, MIRT and HIRT on gains in muscle strength and functional performance. In addition, BFR training and HIRT revealed similar increases in muscle mass, which were not observed with LIRT and MIRT. Notably also, BFR training appears to yield greater gains in muscle strength compared to LIRT in OA patients [37]. The authors concluded that low-intensity BFR training represents a promising rehabilitation strategy, which is a viable alternative or adjunct to resistance training that includes higher loading intensities, exceeding 60% 1RM [37]. However, to date only few BFR studies have been conducted in knee OA patients involving a low total number of participants performing BFR training [25,26,101]. Thus, more studies are needed to clarify the effect of low-intensity BFR training on mechanical lower limb muscle function and functional performance in patients with OA.

Consequently, the present randomized assessor-blinded controlled trial was designed to examine the effect of low-intensity BFR training versus SR on mechanical muscle function, self-reported physical function, and objective measures of functional performance in men and women with radiographically diagnosed knee OA.

## Supporting information

**S1 Checklist.**
(DOCX)

**S2 Checklist.**
(PDF)

## Author Contributions

**Conceptualization:** Brian Sørensen, Per Aagaard, Sofie K. Hansen, Charlotte Suetta, Christian Couppé, S. Peter Magnusson, Finn E. Johannsen.

**Data curation:** Brian Sørensen, Finn E. Johannsen.

**Formal analysis:** Per Aagaard, Sofie K. Hansen, Charlotte Suetta, S. Peter Magnusson, Finn E. Johannsen.

**Funding acquisition:** Brian Sørensen, Sofie K. Hansen, Finn E. Johannsen.

**Investigation:** Brian Sørensen, Sofie K. Hansen, Charlotte Suetta, Finn E. Johannsen.

**Methodology:** Brian Sørensen, Per Aagaard, Mikkel H. Hjortshøj, Sofie K. Hansen, Charlotte Suetta, Christian Couppé, S. Peter Magnusson, Finn E. Johannsen.

**Project administration:** Brian Sørensen, Finn E. Johannsen.

**Resources:** Brian Sørensen, Charlotte Suetta, Finn E. Johannsen.

**Software:** Sofie K. Hansen, S. Peter Magnusson.

**Supervision:** Per Aagaard, Charlotte Suetta, Christian Couppé, S. Peter Magnusson, Finn E. Johannsen.

**Validation:** Brian Sørensen, Per Aagaard, S. Peter Magnusson, Finn E. Johannsen.

**Visualization:** Brian Sørensen.

**Writing – original draft:** Brian Sørensen.

**Writing – review & editing:** Brian Sørensen, Per Aagaard, Mikkel H. Hjortshøj, Sofie K. Hansen, Charlotte Suetta, Christian Couppé, S. Peter Magnusson, Finn E. Johannsen.

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
