## [Decision Letter · Decision Letter 0]

6 Oct 2023

PONE-D-23-14776Physiological and clinical effects of low-intensity blood-flow restricted resistance exercise compared to standard rehabilitation in adults with knee osteoarthritis – protocol for a randomized controlled trialPLOS ONE

Dear Dr. Sørensen

Thank you for submitting your manuscript to PLOS ONE. After careful consideration, we feel that it has merit but does not fully meet PLOS ONE’s publication criteria as it currently stands. Therefore, we invite you to submit a revised version of the manuscript that addresses the points raised during the review process.

ACADEMIC EDITOR

First of all, I want to apologize for the prolonged peer-review time. Securing reviewers has been challenging; therefore, I have also tried to support the available external peer reviews performed. However, this is a very thoroughly described protocol, with only minor revisions necessary, but nonetheless important ones. For your information, you should disregard the comments from reviewer 2 because the peer review performed does not take into account that this is a protocol and not a manuscript reporting a completed trial. 

We look forward to receiving your revised manuscript.

Kind regards,

Daniel Ramskov, Ph.D

Academic Editor

PLOS ONE

Journal Requirements:

Additional Editor Comments (if provided):

**Suggestions.**

-Maybe add a hypothesis, following the aim?

-Consider if TIDieR could support the description and transparency of the intervention even further?

**Comments.**

-Please revise the manuscript gramatical and check spelling.

- Line 123 it is stated that assesment will take place after 24 weeks. Line 260 assesment is reported to take place after 36 weeks, and also in line 445? ClinincalTrials says 6 months (24 weeks).

- Line 263, what standardized test procedures, please report, refer or provide reference for the standardized descriptions.

- Line 327, maybe in bold instead of Italic?

- Regarding the power calculation - please clarify or consider why you calculate the required n, based on within-group differences, when you describe the primary outcome as the between-group difference? It´s a superiority trial is it not? 

- The 10 point difference you report as considered clinical relevant is that within-group or between-group. PLease clarify.

Reviewers' comments:

Reviewer's Responses to Questions

**Comments to the Author**

1. Does the manuscript provide a valid rationale for the proposed study, with clearly identified and justified research questions?

Reviewer #1: Yes

Reviewer #2: Yes

2. Is the protocol technically sound and planned in a manner that will lead to a meaningful outcome and allow testing the stated hypotheses?

Reviewer #1: Yes

Reviewer #2: Partly

3. Is the methodology feasible and described in sufficient detail to allow the work to be replicable?

Reviewer #1: Yes

Reviewer #2: No

4. Have the authors described where all data underlying the findings will be made available when the study is complete?

Reviewer #1: No

Reviewer #2: Yes

5. Is the manuscript presented in an intelligible fashion and written in standard English?

Reviewer #1: Yes

Reviewer #2: No

6. Review Comments to the Author

You may also provide optional suggestions and comments to authors that they might find helpful in planning their study.

Reviewer #1: In this study protocol, a two-arm randomized controlled trial is being proposed to evaluate the effect of low-load blood-flow restriction compared to standard rehabilitation on pain, thigh muscle mass and mechanical muscle function in adults with knee osteoarthritis. The primary outcome variable is the change in KOOS-Pain subscale from baseline to 12 weeks. Secondary outcome variables are changes in pain sensitivity, functional performance, muscle mass and mechanical muscle function. Assessment will be performed at three time points: baseline, after 8 weeks of training and at the end of the intervention period (12 weeks). Mixed linear models will be used for the analysis.

Minor revisions:

1- Line 135: If block randomization will be used, state the block size.

2- Line 455: Indicate the statistical testing method which achieves 80% power.

3- Line 464: Indicate the underlying covariance structure that will be used in the mixed linear model or the criteria for selecting it.

4-Line 471: Clarify if the mixed model or some other statistical method will be used to correlate the primary outcome to the secondary outcomes.

5- The abstract indicates that one-way analysis of variance (ANOVA) will be used for data analysis; however, ANOVA is not mentioned in the “Statistical analysis” section. Clarify.

6- State the software that will be used for the statistical analysis.

Reviewer #2: Topic:

The topic is not focused. The topic should be neither be too short nor too long. In your case it is too lengthy.

It is not necessary to write the type of research (i.e. a randomized controlled trial) b/c it has its own section (in the method and material part).

Abstract Part:

The abstract is not written in an intelligible fashion and in Standard English. For example it says the aim is to examine the effect of. You better to use was. Because this manuscript is completed, it is not a proposal.

The other thing is that almost all the manuscript was written in future tense. You used the future ‘’will be…’’ several times throughout the manuscript. Thus, you should write in Standard English. Because you already submitted a completed manuscript, so why you write in the form of future tense.

I didn’t see the RESULT and CONCLUSION of the study in the abstract part, so why?

Main Part of the Manuscript

Still you didn’t write in Standard English. Thus, use standard writing techniques.

Table-3

Page-12 under the outcome variables, you write as Table-3, however; I didn’t find this table-3 in your manuscript. So why?

Results [Major comments]

You must show the result separately in precise manner.

This is the most important part of the study. However; to my knowledge, you didn’t give attention to it.

You didn’t show your finding precisely in the form of tables. I didn’t see any table that show your results. Because this is an experimental study, it is not a qualitative study which is expressed in texts only.

Discussion [Major comments]

You tried to discuss your study but it is not well-done. You should discuss it critically by including equivocal studies and what you’re finding shows and it’s significant?

7. PLOS authors have the option to publish the peer review history of their article (what does this mean?). If published, this will include your full peer review and any attached files.

Reviewer #1: No

Reviewer #2: No

---

## [Author Response · Author response to Decision Letter 0]

16 Nov 2023

Editor Comments:

(1) Please revise the manuscript gramatical and check spelling.

Author’s reply: 

The revised manuscript has been revised for grammar and spelling.

(2) Line 123 it is stated that assesment will take place after 24 weeks. Line 260 assesment is reported to take place after 36 weeks, and also in line 445? ClinincalTrials says 6 months (24 weeks).

Author’s reply: 

Thank you for pointing out this inconsistency. Sentences have been rephrased accordingly and it is now stated that follow-up questionnaires will be assessed 24 weeks after the intervention period throughout the entire manuscript (p. 6, lines 133-134 and p. 11, lines 270-271: “…completed 24 weeks after the intervention period.”). 

P. 16, line 445 in the original manuscript seems to have been a mistake. The sentence has been rephrased to “The raw data will be stored for five years after completion of the trial with restricted access to the data” (p. 19, lines 486-487 in the revised manuscript).

(3) Line 263, what standardized test procedures, please report, refer or provide reference for the standardized descriptions.

Author’s reply: 

The sentence has been rephrased to clarify the test procedures in question (p. 11, lines 272-273): “…the assessor will be thoroughly trained in performing the tests according to the specific procedures involved in each test modality (elaborated below).”

(4) Line 327, maybe in bold instead of Italic?

Author’s reply: 

We see Editor’s point. The sentence has been changed to bold instead of italic (p. 15, line 340): “Secondary outcomes related to mechanical muscle function”

(5) Regarding the power calculation - please clarify or consider why you calculate the required n, based on within-group differences, when you describe the primary outcome as the between-group difference? It´s a superiority trial is it not? 

Author’s reply: 

We thank Editor for the question and see the inconsistency in the sentence phrase. This is a superiority trial, and the power calculation is based on between-group differences. The sentence has been rephrased accordingly (p. 20, lines 494-499): “The estimated number of participants in the study are based on the primary outcome variable, KOOS-Pain subscale (58), with the assumption that a between-group change of 10 points is considered clinically relevant (58, 60) assuming a standard deviation (SD) of 15 points (11, 60). To achieve a statistical power of 80 % with a significance level of 0.05 and an expected between-group change of 10-points following 12 weeks training (primary endpoint) using a two-sample pooled t test required 37 participants in each group”

(6) The 10 point difference you report as considered clinical relevant is that within-group or between-group. PLease clarify.

Author’s reply: 

Thank you for pointing out this inconsistency. Sentence has been rephrased to clarify (p. 20, line 495): “…with the assumption that a between-group change of 10 points…”

Reviewer #1:

(1) Line 135: If block randomization will be used, state the block size.

Author’s reply: 

Thank you for your question. We are not using block randomization, but as stated a stratified randomization (p. 6, lines 145-149).

(2) Line 455: Indicate the statistical testing method which achieves 80% power.

Author’s reply: 

Thank you for pointing out this missing information. It is now included in the sample size section (p. 20, lines 497-499): “To achieve a statistical power of 80 % with a significance level of 0.05 and an expected between-group change of 10-points following 12 weeks training (primary endpoint) using a two-sample pooled t test required 37 participants in each group”

(3) Line 464: Indicate the underlying covariance structure that will be used in the mixed linear model or the criteria for selecting it.

Author’s reply: 

Thank you for pointing out this inconsistency. Sentence has been rephrased and it is now stated which covariance structure will be used (p. 20, lines 505-506): ”…mixed linear model analysis with an autoregressive covariance structure will be used…”

(4) Line 471: Clarify if the mixed model or some other statistical method will be used to correlate the primary outcome to the secondary outcomes.

Author’s reply: 

Thank you for pointing out this missing information. It has now been included in the statistical analysis section (p. 20, lines 514-516): “The primary outcome will be correlated to the various secondary outcomes using multiple linear regression analyses.”

(5) The abstract indicates that one-way analysis of variance (ANOVA) will be used for data analysis; however, ANOVA is not mentioned in the “Statistical analysis” section. Clarify.

Author’s reply: 

Thank you for pointing out this inconsistency. It is now included in the statistical analysis section (p. 20, lines 511-514): “All outcome parameters (primary, secondary) will be analyzed using a one-way analysis of variance model to analyze between-group mean changes. The model includes changes from baseline to 8- and 12-weeks. Pre-to-post intervention comparisons will be analyzed using a mixed linear model approach”

(6) State the software that will be used for the statistical analysis.

Author’s reply: 

Thank you for pointing out this missing information. It is now stated which statistical software that will be used (p. 20, lines 517-518): “All statistical analyses will be performed in a blinded fashion SPSS (Version 22, IBM SPSS Statistics).”

Reviewer #2:

(1) The topic is not focused. The topic should be neither be too short nor too long. In your case it is too lengthy.

Author’s reply: 

We see and understand reviewer’s point. We have chosen to be very thorough in describing both the topic and the overall project as it is a protocol article.

(2) It is not necessary to write the type of research (i.e. a randomized controlled trial) b/c it has its own section (in the method and material part).

Author’s reply: 

We understand reviewer’s point. However, we have chosen to be clear in the lengthened title that this is a protocol for a randomized controlled trial.

(3) The abstract is not written in an intelligible fashion and in Standard English. For example it says the aim is to examine the effect of. You better to use was. Because this manuscript is completed, it is not a proposal. The other thing is that almost all the manuscript was written in future tense. You used the future ‘’will be…’’ several times throughout the manuscript. Thus, you should write in Standard English. Because you already submitted a completed manuscript, so why you write in the form of future tense.

Author’s reply: 

We understand reviewer’s point. We would use past tense in a manuscript with a completed study. However, we are still collecting data and therefore writing in future tense.

(4) I didn’t see the RESULT and CONCLUSION of the study in the abstract part, so why?

Author’s reply: 

We understand reviewer’s question. This is a manuscript for a protocol article why we have no data to publish, and therefore cannot conclude anything yet.

(5) Table-3 Page-12 under the outcome variables, you write as Table-3, however; I didn’t find this table-3 in your manuscript. So why?

Author’s reply: 

Thank you for pointing out this inconsistency and missing table. There is no Table 3, but Table 2 is missing. The sentence has been rephrased and Table 2 has been included in the manuscript (p. 13, lines 282-283).

---

## [Editor Report · Decision Letter 1]

28 Nov 2023

Physiological and clinical effects of low-intensity blood-flow restricted resistance exercise compared to standard rehabilitation in adults with knee osteoarthritis – protocol for a randomized controlled trial

PONE-D-23-14776R1

Dear Dr. Sørensen,

We’re pleased to inform you that your manuscript has been judged scientifically suitable for publication and will be formally accepted for publication once it meets all outstanding technical requirements.

Kind regards,

Daniel Ramskov, Ph.D

Academic Editor

PLOS ONE
---

## [Editor Report · Acceptance letter]

5 Dec 2023

PONE-D-23-14776R1 

Physiological and clinical effects of low-intensity blood-flow restricted resistance exercise compared to standard rehabilitation in adults with knee osteoarthritis – protocol for a randomized controlled trial 

Dear Dr. Sørensen:

I'm pleased to inform you that your manuscript has been deemed suitable for publication in PLOS ONE. Congratulations! Your manuscript is now with our production department. 

Kind regards, 

on behalf of

Dr. Daniel Ramskov 

Academic Editor

PLOS ONE